

# Microplastic pollution at Qilianyu, the largest green sea turtle nesting grounds in the northern South China Sea

Ting Zhang[1,*], Liu Lin[1,*], Deqin Li[1], Jichao Wang[1], Yunteng Liu[2], Rui Li[1], Shannan Wu[1] and Haitao Shi[1]

[1] Ministry of Education Key Laboratory for Ecology of Tropical Islands, Key Laboratory of Tropical Animal and Plant Ecology of Hainan Province, College of Life Sciences, Hainan Normal University, Haikou, China
[2] Marine Protected Area Administration of Sansha City, Sansha, China
* These authors contributed equally to this work.

## ABSTRACT

Microplastics, new persistent pollutants, have recently attracted considerable attention. When present in beach sediments, microplastics may adversely affect the nesting and hatching of sea turtles on beaches. In this study, we investigate microplastic pollution at Qilianyu (northeastern Xisha Islands), the largest known nesting ground for green sea turtles (*Chelonia mydas*) in China. We found that the average abundance of microplastics in the beach surface sediments was 338.44 ± 315.69 thousand pieces·m$^{-3}$ or 1,353.78 ± 853.68 pieces·m$^{-2}$, with foam and fragments as the main microplastic type identified. The microplastic particles were categorised as small and were predominantly within the 0.05–1 mm size category. Most microplastic particles were white (71.31%). Polystyrene and polyethylene were found to be the most common forms of plastic present. Microplastic pollution was not only observed on the beach surface but also at the bottom of nests approximately 60 cm may be harmful to the incubation of sea turtle eggs. We suggest removing plastic litter, especially small pieces of plastic, on beaches to reduce the threat of microplastic pollution to marine life, including sea turtles. Furthermore, the foam used in aquaculture should be recovered and replaced before it becomes fragmented due to age. In addition, regional cooperation between stakeholders in the South China Sea should be strengthened to collectively promote the reduction and cleanup of marine litter.

## INTRODUCTION

Marine plastic pollution is a common worldwide environmental problem (*Martin et al., 2019*). Plastic fragments in the ocean are easily broken down into many miniature fragments or particles as a result of long-term physical and chemical action (*Hopewell, Dvorak & Kosior, 2009*; *Andrady, 2011*). Microplastics are defined as plastic fragments or particles that have a diameter of less than 5 mm (*Andrady, 2011*). It is estimated that

Corresponding author
Haitao Shi, haitao-shi@263.net

approximately 51 trillion plastic particles are present in the global oceans (*Wessel et al., 2016*; *Tirkey & Upadhyay, 2021*). Microplastics in the environment can migrate over long distances through external forces such as wind, rivers, and ocean currents (*Claessens et al., 2011*). They can pollute some of the most remote corners of the earth, from mountain lakes to deep-sea sediments (*Moreira et al., 2016*; *Nelms et al., 2017*; *Gago et al., 2018*). Therefore, although microplastics are a new pollutant in the current marine environment, they are gaining increasing global attention (*Andrady, 2011*). However, most studies focus on offshore waters, with limited research focusing on remote areas such as deep seas, polar regions, islands, and reefs (*Auta, Emenike & Fauziah, 2017*; *Imhof et al., 2013*).

Sea turtles are umbrella species of the marine ecosystem and flagship species for marine conservation (*Bouchard & Bjorndal, 2000*; *Hamann et al., 2010*). There are seven species of sea turtles worldwide, with five species in China, of which the green sea turtle is the most abundant (*Chan et al., 2007*). Sea turtles are highly loyal to their nesting grounds, with most adult females returning to the beach where they were born to lay their eggs (*Triessnig, Roetzer & Stachowitsch, 2012*). The presence of microplastics on beaches can have a negative effect on the reproductive cycle of sea turtles (*Beckwith & Fuentes, 2018*; *Duncan et al., 2018*). Sea turtles have temperature-dependent sex determination, indicating that the sex of offspring depends on the incubation temperature. The mid-stage of the incubation period is the pivotal period for sex determination (*Mrosovsky & Yntema, 1980*). The specific heat capacity of plastics is higher than that of sand, so microplastics incorporated into beach sand will increase the overall temperature of the beach (*Andrady, 2011*). This phenomenon will affect the nest temperature and cause a gender imbalance in sea turtles (*Beckwith & Fuentes, 2018*). Furthermore, microplastics can absorb and later release harmful chemical pollutants that can into the egg through migration and can affect embryonic development and decrease hatching success (*Bergeron, Crews & McLachlan, 1994*; *Yang et al., 2011*; *Jian et al., 2021*). *Duncan et al. (2018)* first discovered microplastics at a nest depth of 60 cm in Mediterranean Cyprus, and raised the possibility that the presence of microplastics affect the hatching success rate and sex ratio of sea turtles, threatening their population sustainability.

The populations of sea turtles in China have dropped sharply owing to massive illegal trade and habitat loss (of almost all nesting beaches) (*Lin et al., 2021*). The Xisha Islands in the South China Sea are currently the largest nesting grounds for green sea turtles (*Chelonia mydas*) in China. A total of 100 *C. mydas* nests were recorded each year from 2016 to 2019 (*Jia et al., 2019*; *Wang et al., 2019*). These islands are the last surviving, relatively intact land area in China where sea turtle reproduction still occurs, mainly due to the remoteness of the islands, *i.e.*, distance from the mainland and low human population due to low levels of development. The green sea turtle population in the Xisha Islands has a unique genetic makeup, and represents a newly defined population (*Gaillard et al., 2020*). It is therefore vital to protect this distinct turtle population and their nesting grounds to ensure their survival into the future.

In this study, the abundance of microplastics in the nesting grounds of *C. mydas* at Qilianyu, northeastern Xisha Islands is evaluated. The characteristics and possible sources of microplastic pollutants are also described in this region. We propose revised management practices in line with the survey results. These data will help fill gaps in the knowledge regarding microplastic pollution in *C. mydas* habitats in China, and will provide basic information and references for the protection, management, and ecological restoration of beaches as sea turtle nesting grounds in the South China Sea.

## MATERIALS AND METHODS

### Study area

Qilianyu is a subgroup of islands is located in the northeastern Xisha Islands in the South China Sea (16°55′N–17°00′N, 112°12′E–112°21′E), approximately 330 km from Hainan Island. The eight small islands are connected by reefs, with a total area of approximately 1.32 km$^2$. With the exception of Zhaoshu Island and North Island, all other islands in the archipelago are uninhabited. The current permanent human population is approximately 200, primarily living on Zhaoshu Island. In this study, sampling points were set up on six islands, including North Island (NI), Middle Island (MI), South Island (SI), North Sand (NS), Middle Sand (MS), and South Sand (SS). These islands have good quality nesting sites and have recent records of *C. mydas* nesting (*Zhang et al., 2020*) (Fig. 1).

### Sample collection and separation

The geographic coordinates of the sample points were recorded using a global positioning system. Sediment samples were collected within an area of 25 cm × 25 cm and a depth of 0–2 cm from both the strand line and the turtle nesting line (TNL) for six nesting grounds. This sampling process was repeated three times at each nesting ground. Additional samples were collected from the TNL using a custom-made cylindrical galvanised steel core with a diameter of 20 cm. Samples were collected at depths of 0–60 cm (0–2, 2–20, 21–40, and 41–60 cm) to explore the presence and extent of microplastic pollution at the sea turtle nesting depth of approximately 60 cm (*Duncan et al., 2018*).

The saturated sodium chloride density method described by *Zhang et al. (2021)* was used to separate microplastics from the sediment. For each sample, 250 cm$^3$ of sand was placed in a beaker. A total of 500 mL of saturated sodium chloride solution (1.2 g·mL$^{-1}$) was added. The mixture was then stirred for 2 min, after which it was left to settle for 10 min. The supernatant was then passed through a 300-micron mesh sieve. The remaining solids in the beaker were added to a sodium iodide solution (1.8 g·mL$^{-1}$) and stirred for 2 min, after which the mixture was left to settle for 10 min. After density separation had taken place, the sample was transferred to a 100 mL beaker. A solution of 10% potassium hydroxide was added, and the mixture was left to digest for 2 days. Finally, the supernatant solution was decanted and filtered through a 0.45 μm glass fiber membrane (GF/F, 47 mm Ø; Whatman, Shanghai, China). This was done using a vacuum filtration device (GM-0.33A, Zhengzhou, China), while waiting for the one-step analysis (*Thompson et al., 2004*; *Wang et al., 2018*).
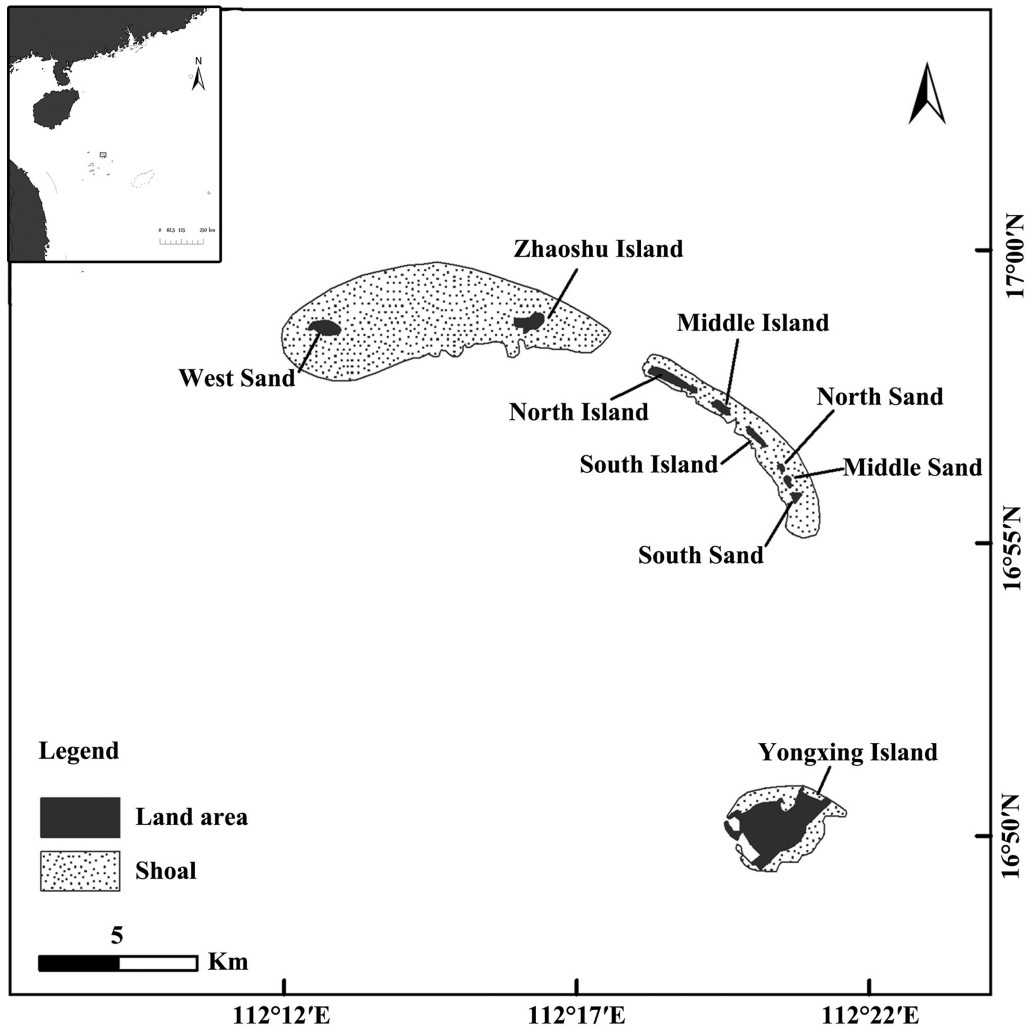

**Figure 1 Map of study area and sampling points.** Sampling points: NI, North Island; MI, Middle Island; SI, South Island; NS, North Sand; MS, Middle Sand; South Sand (SS).

## Observation and identification of microplastics

All samples on the filter membrane were observed under a stereo microscope (SMZ-168 SERIES; MOTIC, Xiamen, China), and images were obtained with a SONY DSC-RX10M2 digital camera. The microplastics were classified and counted according to their morphological characteristics, colour, and size (*Zhang et al., 2021*).

Samples suspected to be microplastics that were representative of each group were randomly selected, and their surface structures were tested for polymer types using a Fourier transform infrared spectrophotometer (IRTracer-100; SHIMADZU, Kyoto, Japan). The detector spectral range was 600–4,000 cm$^{-1}$, co-adding 16 scans at a resolution of 8 cm$^{-1}$ (*Zhang et al., 2021*). The resulting atlas was compared to the IR polymer spectral library (ATR-Polymer2, IRs Polymer2, T-Polymer2, and Shimadzu Standard Library),

with only readings at a confidence level of 70% or higher being considered reliable and accepted.

## Experiment quality control

All containers were rinsed at least three times with Milli-Q water and then dried before the start of the experiments. All plastic equipment was replaced with non-plastic ones if possible. If this was not possible, the equipment were rinsed three times with Milli-Q water and then inspected to ensure that no plastic fragments were generated during sample processing. In addition, all containers were always covered with aluminium foil to avoid contamination. Nitrile gloves and cotton lab coats were worn throughout the experiment, with laboratory windows remaining closed. Three procedural blanks were set to minimise contamination from the environment, and results showed that no microplastic particles were detected.

## Statistical analysis

Statistical analysis was performed using Excel and SPSS 19.0 statistical software. All data were tested for normal distribution and variance homogeneity before the statistical analysis. One-way analysis of variance was used to analyse the difference in microplastic abundance between the six nesting grounds. The relevant data are shown as mean ± standard deviation. $P < 0.05$ was considered a significant difference, and $P < 0.01$ was considered a highly significant difference according to the two-tailed test.

# RESULTS

## Distribution and abundance of microplastics pollution at Qilianyu

The quantity of microplastics found in the nesting grounds at Qilianyu ranged as 92–782 thousand pieces·m$^{-3}$ or 368–3,128 pieces·m$^{-2}$, with an average abundance of 338.44 ± 315.69 thousand pieces·m$^{-3}$ or 1,353.78 ± 853.68 pieces·m$^{-2}$. The distribution of microplastics across the six islands showed a degree of spatial variation (Fig. 2). MS was the nesting ground that was most severely polluted with microplastics, followed by NS, SI, and NI, respectively ($df = 17$; $F = 7.202$; $P = 0.002$). In contrast, MI and SS were less polluted than the other sampling sites. The abundance of microplastics in the sediment samples exhibited a gradual increase from northwest to southeast, with the exception of MI and SS.

When comparing the abundance of beach microplastics with other areas (Table S1), the abundance of microplastics (0.05–5 mm in size) in nesting grounds of *C. mydas* at Qilianyu was lower than in Hainan Island, Hong Kong, and Guangdong Province, but similar to that in Ganquan and Quanfu Island in the Xisha Islands. Moreover, microplastics with a particle size range of 0.05–0.33 mm accounted for 27.79% of all particles in this study. The actual abundance of microplastics at Qilianyu is therefore considerably lower than that found in Guangdong (0.315–5 mm) and Hong Kong (0.315–5 mm).

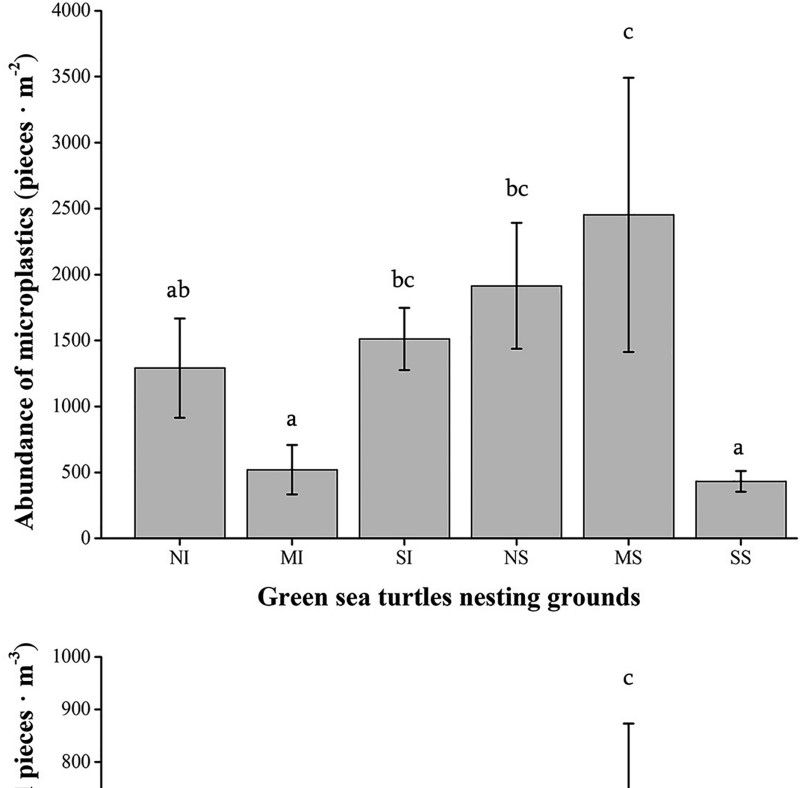

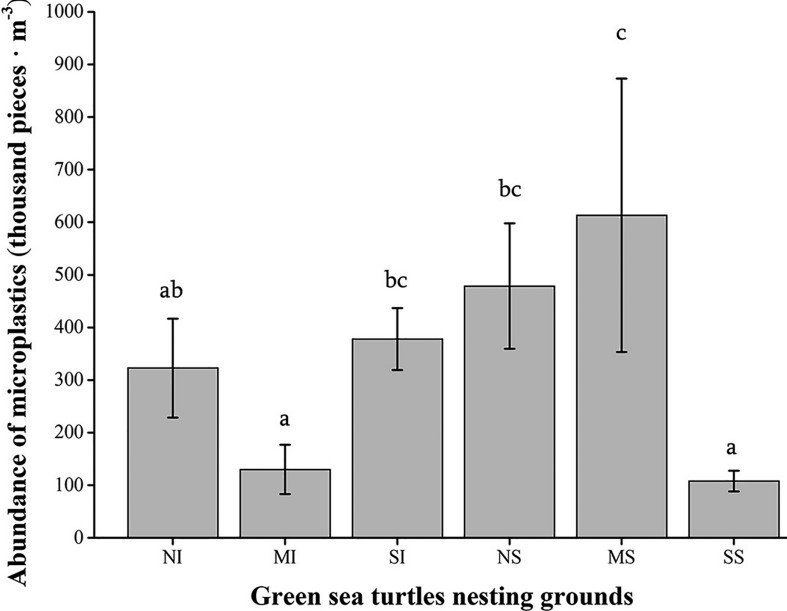

**Figure 2 Microplastic abundance in surface sediments of six nesting grounds at Qilianyu.** Different lowercase letters in the figure indicate significant differences at the $P < 0.05$ level. Notes: NI, North Island; MI, Middle Island; SI, South Island; NS, North Sand; MS, Middle Sand; SS, South Sand.

## Morphological characteristics of microplastics

When microplastics were separated, they were shown to have different morphological characteristics. Figure S1 shows the shape categories of the microplastics found in the samples. Among the microparticles observed, fragments formed the largest proportion at

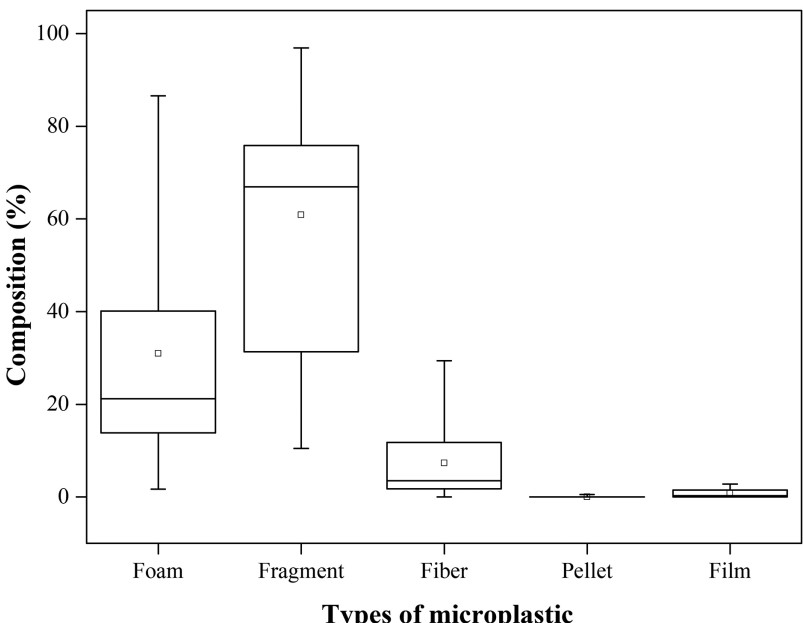

**Figure 3 Composition (%) of microplastics with different shapes (*n* = 18).** The solid horizontal lines from the top to the bottom of each box plot indicate the maximum value, 75% quartile, median, 25% quartile. Note: *n* = 18 is referring to the number of samples.

60.85%, followed by foams at 30.98% and fibers at 7.35%. Meanwhile, pellets and films were relatively rare, accounting for only 0.41% of the total microplastic particles (Fig. 3).

The most common colour of the sampled microplastics was white (71.31%), which included both transparent and white particles (Fig. 4). Among this colour category, white foam was the most common type. The second most common colour was black (23.00%), Multicoloured microplastics such as yellow, green, grey, and blue were relatively rare. The average size of the microplastics at Qilianyu is indicated in Fig. 5. Small microplastic particles (<1 mm) comprised the majority of the microplastics (87.18%).

## Polymer compositions of microplastics

The polymer compositions of the microplastics included polyethylene (PE), polypropylene (PP), and polystyrene (PS) (Fig. S2). The most common polymer compositions were PS (40.74%) and PE (40.74%). PS was found in foam, while PE found in fibers, pellets, and fragments, while PP was found in fragments (Fig. 6).

## Changes in microplastic density with increasing sampling depth

As the sampling depth increased, the average density of the microplastics decreased (Fig. 7). However, there was no significant difference between microplastic densities at each depth ($df$ = 66; $F$ = 2.043; $P$ = 0.117 > 0.05) (Table S2). This indicates that microplastic pollution was not limited to the beach surface, and that that microplastics can come into close contact with the turtle eggs, which usually lie at a depth of 60 cm.

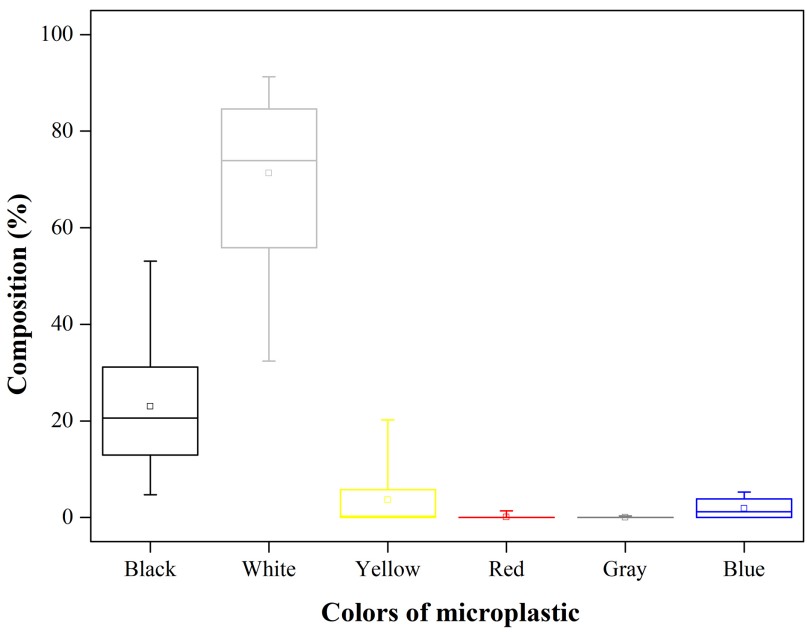

**Figure 4 Composition (%) of microplastics with different colors (*n* = 18).** The solid horizontal lines from the top to the bottom of each box plot indicate the maximum value, 75% quartile, median, 25% quartile. Note: *n* = 18 is referring to the number of samples.

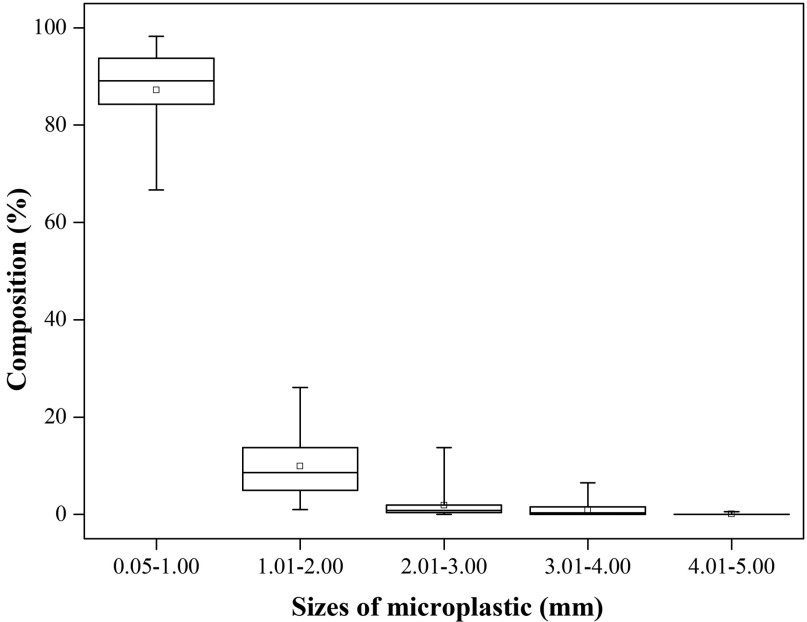

**Figure 5 Composition (%) of microplastics with different grain sizes (*n* = 18).** The solid horizontal lines from the top to the bottom of each box plot indicate the maximum value, 75% quartile, median, 25% quartile. Note: *n* = 18 is referring to the number of samples.

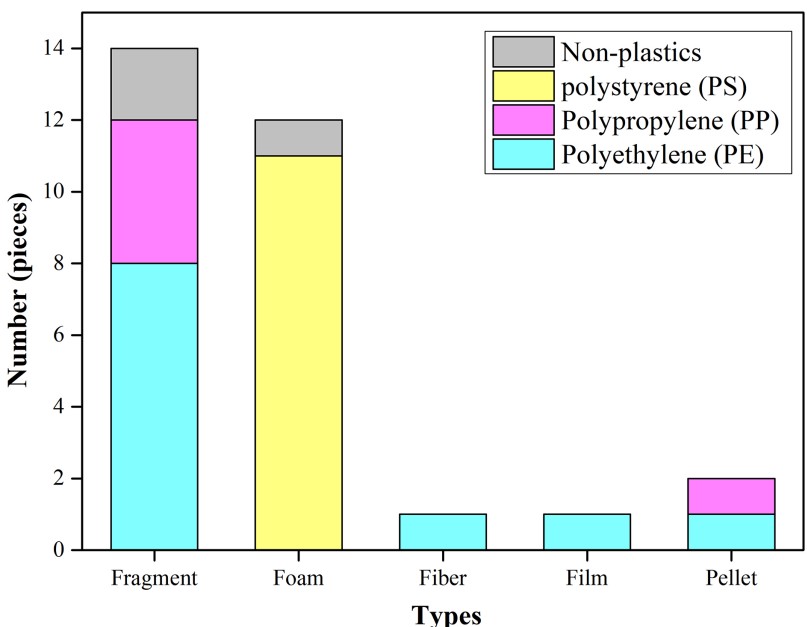

**Figure 6 Composition of the selected items from six nesting grounds at Qilianyu.**

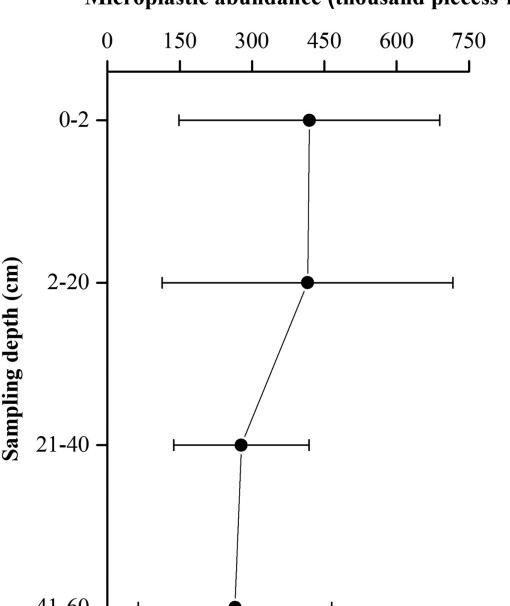

**Figure 7 Values of average microplastic abundance at different depths in the nesting grounds of green sea turtles.**

## DISCUSSION

### Current status of microplastic pollution at Qilianyu

Beaches are gathering points for ocean microplastics and key areas of environmental pollution (*Poeta, Battisti & Acosta, 2014*; *Nelms et al., 2016*). Although Qilianyu is

relatively remote from the mainland, the nesting grounds here for *C. mydas* are affected by microplastic pollution. Although many studies of beach microplastic pollution exist, the particle size survey range obtained was non-uniform, making it difficult to compare the variation in abundance of microplastics at the regional level. Therefore, only broad comparisons can be made for the same particle sizes across different studies (*Fang et al., 2021*). The overall abundance of microplastics (size range 0.05–5 mm) at Qilianyu (1,353.78 ± 853.68 pieces·m$^{-2}$) was lower than that at other areas such as Hainan (5,595 pieces·m$^{-2}$) and Guangdong (6,675 ± 7,021 pieces·m$^{-2}$), China (*Fok et al., 2017*; *Zhang et al., 2021*). Microplastic pollution is closely related to regional population activities and economic development (*Fang et al., 2021*). It is likely that the lower abundance of microplastics at Qilianyu in the Xisha Islands is associated with increasing distance from the mainland and the small human population of the islands. However, microplastics are stable and can exist in the environment for a long time, and their abundance may increase with time. Therefore, measures must be taken to prevent the microplastic pollution increase at Qilianyu.

## Sources of microplastic pollution for Qilianyu

The types of microplastics found at the nesting grounds at Qilianyu were primarily fragments and foams, with the main compositions being PS and PE. For comparison, *Huang et al. (2020)* showed that the types of microplastics found in the sea water of the Xisha Islands are primarily fibers and films, with the predominant composition being PET (56.2%) and PP (20.3%). The different types and compositions found indicate that microplastics in beach sediment at Qilianyu may not come directly from local sea water but primarily from the decomposition of beach litter.

A previous study on the general categorisation of beach litter (size > 1 cm) at Qilianyu found that the greatest proportion was plastic and foam (*Zhang et al., 2020*) (Fig. S3). Fragments and foam can easily breakdown in a beach environment, due to higher temperatures increasing weathering and degradation (*Fok & Cheung, 2015*; *Fok et al., 2017*). Owing to its tropical climate, Qilianyu experiences strong direct solar radiation, which accounts for 60–70% of the global solar radiation (*Ye, 1996*). The average annual temperature is approximately 27.4 °C, which is conducive to the breakdown of plastic (*Xu et al., 2018*). Furthermore, the highest percentage of the microplastics at Qilianyu were white, followed by black. In line with the results of 24 investigations analysed by *Hidalgo-Ruz et al. (2012)*, this trend may be due to the weathering and fading of plastics in beach or ocean environments. Therefore, it is likely that most of the microplastics found in this study were from broken plastic litter on the beach. Items such as plastic bottle caps being broken into small plastic particles on the beach was commonly observed during field work (Fig. S4).

## Potential threats of microplastic pollution to sea turtles at Qilianyu

Consistent with previous research results (*Vianello et al., 2013*; *Mohamed Nor & Obbard, 2014*; *Peng et al., 2017*), the microplastics found in the nesting grounds on Qilianyu were predominantly small particles (0.05–1 mm). However, the smaller the microplastic

particle size, the larger their specific surface area. This implies that the microplastics can absorb more pollutants, which may later be released and cause greater harm to the hatching of green sea turtles (*Duncan et al., 2018*).

The presence of microplastics is extensive in the sea turtle nesting grounds at Qilianyu, with close contact between microplastic and eggs. Owing to the effects of climate change and the presence of microplastics, the beach temperature at Qilianyu has increased annually. Beach temperatures have increased by 1–2 °C in 2021 (the average temperature was 32.4 °C) as compared to data collected in 2018 (the average temperature was 30.2 °C) (T. Zhang & L. Lin, 2021, unpublished data). An increased incubation temperature may change the sex ratio of sea turtles hatched locally. In addition, the microplastic surfaces can accumulate and release heavy metals and organic pollutants (*Bergeron, Crews & McLachlan, 1994*; *Yang et al., 2011*). *Jian et al. (2021)* indicated that heavy metals can enter the embryo by penetrating the shell membrane. Therefore, we suggest that microplastics near the *C. mydas* nests may adversely affect the development of turtle embryos. However, the degree of harm to the hatching of *C. mydas* eggs due to microplastics at Qilianyu remains unclear. Therefore, research and field monitoring must be strengthened. Important areas for future research include determining the impact of microplastic enrichment on turtle hatching temperature, and the impact of attached microplastic surface pollutants on sea turtle hatching.

## Management suggestions

*Fok et al. (2017)* suggested that cleaning up plastic litter on beaches may reduce the generation of microplastics there. The current beach litter cleaning at Qilianyu primarily removes large plastic litter >10 cm. The overall amount of large litter has been reduced to low levels by regular cleaning efforts. However, a large amount of small litter (1–10 cm) still remains after cleaning has taken place. Thus, the removal proportion of smaller pieces of litter requires improvement (*Zhang et al., 2020*). In addition, during the peak period for sea turtle nesting, the accumulation rate of plastic litter on the *C. mydas* nesting ground beach at North Island of Qilianyu was 0.47 pieces·m$^{-2}$·month$^{-1}$ or 9.2 g·m$^{-2}$·month$^{-1}$ (D. Q. Li & L. Lin, 2021, unpublished data). This was higher than the average accumulation rate of 2019 Bulletin on the State of China's Marine Ecological Environment (0.58 pieces·m$^{-2}$·two month$^{-1}$ or 10.5 g·m$^{-2}$·two month$^{-1}$) (*Ministry of Ecology & Environment of the People's Republic of China, 2019*). This plastic litter should be cleaned up in a timely manner to prevent its breakdown into form microplastics. Therefore, it is suggested that the strength and frequency of beach litter cleaning should increase from once a week to once every 2–3 days, and focus on the removal of small plastic particles and foam. Considering the increasing number of foam plastics, we suggest that the foam used for local commercial activities such as aquaculture and seafood transportation should be recovered and replaced before aging and fragmenting into smaller plastic pieces.

As the geographic source of large beach litter at Qilianyu was primarily from Southeast Asian countries, such as Vietnam and Malaysia (*Zhang et al., 2020*). Regional cooperation between stakeholders in the South China Sea should be strengthened, with

joint measures on the appropriate treatment of marine litter. This will help to reduce the generation of large plastic litter and prevent breakdown of large plastics to form microplastics, and benefit the conservation of *C. mydas* populations of the South China Sea.

## ACKNOWLEDGEMENTS

We are sincerely grateful for the help of the Sansha Ocean and Fisheries Administration and the Qilianyu Administration Committee.

### Funding

This work was funded by the National Natural Science Foundation of China (31960101, 32170532, 32160135), the Hainan Natural Science Foundation (319MS048) and the Innovative Research Projects for Postgraduate of Hainan Province (Qhyb2021-53). The funders had no role in study design, data collection and analysis, decision to publish, or preparation of the manuscript.

### Grant Disclosures

The following grant information was disclosed by the authors:
National Natural Science Foundation of China: 31960101, 32170532, 32160135.
Hainan Natural Science Foundation: 319MS048.
Innovative Research Projects for Postgraduate of Hainan Province: Qhyb2021-53.

### Competing Interests

The authors declare that they have no competing interests.

### Author Contributions

- Ting Zhang conceived and designed the experiments, performed the experiments, analyzed the data, prepared figures and/or tables, authored or reviewed drafts of the article, and approved the final draft.
- Liu Lin conceived and designed the experiments, performed the experiments, analyzed the data, prepared figures and/or tables, authored or reviewed drafts of the article, and approved the final draft.
- Deqin Li conceived and designed the experiments, authored or reviewed drafts of the article, and approved the final draft.
- Jichao Wang analyzed the data, authored or reviewed drafts of the article, and approved the final draft.
- Yunteng Liu performed the experiments, prepared figures and/or tables, and approved the final draft.
- Rui Li performed the experiments, prepared figures and/or tables, and approved the final draft.
- Shannan Wu performed the experiments, prepared figures and/or tables, and approved the final draft.

- Haitao Shi conceived and designed the experiments, authored or reviewed drafts of the article, and approved the final draft.

## Data Availability

The raw measurements are available in the Supplemental Files.

## Supplemental Information

Supplemental information for this article can be found online at http://dx.doi.org/10.7717/peerj.13536#supplemental-information.

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
