# Peer review of "Microplastic pollution at Qilianyu, the largest green sea turtle nesting grounds in the northern South China Sea"

_PeerJ, doi:10.7717/peerj.13536_

## Round 0.1 · original submission · Minor Revisions

All three reviewers have minor revisions and constructive comments directed at improving the manuscript, please address these.

·

Basic reporting

English was good but could be tightened up--I have provided edits to improve this. Everything else meets standards

Experimental design

no comment

Validity of the findings

no comment

Additional comments

This paper studies the plastic pollution of Qilianyu, China, an important nesting ground of the green sea turtle. This paper not only quantifies the plastic pollution at nesting grounds, but also highlights the potential problems and provides suggestions of solutions.

The basic reporting, experimental design, and validity of findings meet the standards of the journal.

The study is straightforward and solidly done. The only suggestions I have are of the text. The text could be tighter (cutting out a some unnecessary text), and explanations need to be more clear. i have included a pdf with edit suggestions. Please look at the attached file for details.

Reviewer 2 ·

Basic reporting

Abstract: Consider changing the terminology you use to refer to plastic types. What you call plastic blocks are more commonly referred to as fragments. There is also a difference between a microbead and a pellet (also called nurdle, or pre production pellet). I cant tell what the item in figure S1 box e is, because there is no size included. But refer to [Lusher, Amy L., et al. "Is it or isn't it: the importance of visual classification in microplastic characterization." Applied spectroscopy 74.9 (2020): 1139-1153.] for their classification of microplastics. It can be important to differentiate between microbead and pellets because they come from very different sources in the environment.

Line 160: Figure S1- what are the size of these plastic pieces? Please tell us how large each of the red boxes are, or include a ruler or some other measurement.

Line 193: Insert “due to” after the word likely. “This is likely due to the small population of the islands, implying that it is less severely affected by land-sourced plastic litter.

Line 222: This would be a good place to mention the density of the polymers youre seeing on the beach- PP, PE, PS, that are all less dense than seawater and thus float. Reference Brignac, Kayla C., et al. "Marine debris polymers on main Hawaiian Island beaches, sea surface, and seafloor." Environmental science & technology 53.21 (2019): 12218-12226.

Figure S3 and S4: Were larger items found on the beach included in this study? For example, Figure S4 shows a bottle pump that is fragmenting into smaller pieces and Figure S3 shows plastic bottles and a large piece of Styrofoam. Were these included or is it only microplastics?

Original data- thank you for providing the raw data, this is very important! However, in the second table, Column D, the sign is wrong. It currently says <1-2, which I take to mean pieces less than one up to 2. I believe it should be >1-2, meaning greater than 1 to 2. You also need units, is this mm? or cm?

Figure 1: Add abbreviation for each sampling site so the next figure is easier to understand. Example: North Island (NI)

Figure 3 - 5- I cannot tell the difference between the solid circle and the solid diamond. Can you please change one to help differentiate them? There also appears to be an * in some of the box plots, what are these?

Figures 3-5: clarify that the n = 18 is referring to the number of beaches.

Figure 7: Again, the <20-40 cm is misleading to me.

Experimental design

Line 101: What was the density of the sodium chloride solution?

Line 104: What was the density of the sodium iodide solution?

Line 119: What libraries specifically were used? Commercial or in-house library? How many polymers are in this library?

Line 128: Where are the results from the procedural blanks? Were any microplastics found in the blanks?

Validity of the findings

No Comment

Additional comments

Overall, this is a great paper providing baseline data of microplastic concentrations on a sea turtle nesting beach. Some basic reporting data is missing (units on graphs, density of separation solutions, etc), but otherwise is well written. It is important to provide additional information on how spectral analyses were conducted. Your spectra identification of polymers is only as good as the libraries you are searching through. Additional checking should be done manually to ensure correct identification.

Reviewer 3 ·

Basic reporting

The authors explore the topic of microplastic interactions with sea turtle hatchings. Microplastics are an emergent pollutant that has widespread effects. This is an interesting topic worth examining. Overall the manuscript is straightforward and well written. However there are some portions which require clarification and referencing. The differentiation between results and discussion have to be more distinct.

Experimental design

no comment

Validity of the findings

Multiple references are missing throughout the manuscript, especially in the discussion

Additional comments

I have a few specific comments:

Line 20- what was the percentage of white particles? Kindly include percentage.

Line 39- references missing

Line 46- abrupt introduction to sea turtles, which species are you looking at? Would be better to give a short introduction to the sea turtle before describing their characteristics.

Line 53-54- reference missing

Line 56- reference missing

Line 57- the explanation using osmosis is unclear. Osmosis usually involves water. Kindly clarify.

Line 100- Can the authors provide measurement units in terms of microplastics/kg of sand? What was the weight of the sand?

Lines 146-149- should shift to discussion.

Lines 155-157- also shift to discussion

Section 3.4- The multiple values reported in the sentences should be tabulated for easier reading.

Lines 192-193- kindly substantiate the statement. Maybe the authors can consider adding a table with the comparison of microplastics with other neighbouring regions.

Lines 209,211- missing references

Line 220- it is difficult to confirm that the microplastics originated from those countries as no specific experiment was conducted

---

## Round 0.2 · accepted · Accept

The comments of all three reviewers, which were relatively minor, have been addressed and I recommend the paper be accepted for publication in PeerJ.